# Cascade switching current detectors based on arrays of Josephson junctions

Roger Cattaneo [1], Artemii E. Efimov [1,2], Kirill I. Shiianov [1], Oliver Kieler[3] & Vladimir M. Krasnov [1] ✉

Cascade multiplication is widely used to enhance photon detector sensitivity. While vacuum tube and semiconductor photomultipliers achieve high gains in the optical range, their performance at lower frequencies is limited by large work functions. Superconducting detectors overcome this constraint, enabling operation in the terahertz (THz) and microwave (MW) ranges. Here we introduce a concept of cascade-amplified superconducting detectors based on Josephson junction arrays. Interjunction coupling in an array triggers avalanche-like switching of multiple junctions upon photon absorption, resulting in cascade amplification of the readout voltage and an increased signal-to-noise ratio. We present prototypes using either low-$T_c$ linear Nb/Nb$_x$Si$_{1-x}$/Nb arrays or Bi$_2$Sr$_2$CaCu$_2$O$_{8+\delta}$ high-$T_c$ stacked intrinsic Josephson junctions. Both MW and THz responses are analyzed and the advantages of the cascade detector over a conventional single-junction detector are demonstrated. Our findings suggest that Josephson junction arrays hold promise for the development of highly sensitive, broadband MW-to-THz detectors.

Detectors in MW and THz ranges find diverse applications, including security, environmental monitoring, medical imaging, chemical analysis, future telecommunication, and fundamental research[1–3]. Single-photon and photon-counting detectors are essential for quantum optics and electronics[3–6]. However, despite two decades of intense research[1–23], MW and THz single-photon detectors are still not commercially available due to various technical challenges and materials limitations.

The primary challenge arises from the small photon energy, placing constraints on materials. Conventional vacuum tube and semiconductor photodetectors[3] are not suitable for this range. Low-gap materials such as superconductors[4–16], half-metals[12,17–19], and gap-engineered quantum dots[20,21] are required. In addition, in situ gap tuning is necessary to optimize detector performance and adjust the dynamic range[15]. This can be achieved through electrostatic gating[5,11,12,17–21], magnetic field[4,6,15,20], bias[6,9,10,15,22,23], temperature[7–10,15,16,22,23], etc.

Another problem is associated with the large wavelength $\lambda_0$ ~mm, which makes it difficult catching photons by micron-size detectors. A sensitive detector must have high optical absorption efficiency $\chi$, but

size mismatch leads to a large impedance mismatch with free space, reducing $\chi$[24]. Dedicated pickup antennas with sizes ~$\lambda_0$ must be implemented to achieve impedance matching and optimal $\chi \simeq 0.5$[24,25].

The low photon energy imposes stringent requirements on detector characteristics. For single-photon resolution, the Noise-Equivalent Power (NEP) should be in the zW/Hz$^{1/2}$ range[11,12,16,20,21]. To avoid intense background radiation, such devices should be cooled to low temperatures, enabling utilization of superconductors. Superconductivity is beneficial for ultrasensitive detectors due to the absence of Jonson-Nyquist noise in the electrodes. Several existing superconducting detectors, particularly those based on qubits[4–6,11,12], approach the quantum limit of sensitivity. The base element exhibiting quantum-mechanical behavior[26], is the Josephson junction (JJ).

A current-biased JJ can function as a sensitive switching current detector (SCD) in a broad MW-to-THz range[15,22,23,27–34]. The responsivity of SCD can be very high, ultimately limited by quantum fluctuations[15,34] or phase diffusion[15,35]. The upper frequency depends on the characteristic voltage, $V_c$, and ranges from sub-THz for low-$T_c$, to THz-range for high-$T_c$ JJs[29,31,32,36]. The highest $V_c \gtrsim 30$ mV[37,38] is

[1]Department of Physics, Stockholm University, AlbaNova University Center, Stockholm, Sweden. [2]Department of Physics, University of Basel, Basel, Switzerland. [3]Physikalisch-Technische Bundesanstalt, Braunschweig, Germany. ✉e-mail: Vladimir.Krasnov@fysik.su.se

achieved in intrinsic Josephson junctions (IJJ) naturally formed in Bi-2212 cuprates[39]. The operation of IJJs above 10 THz has been reported[29,36]. The atomic scale of IJJs leads to a strong mutual coupling and enables coherent behavior[29,32,40–45]. The coupling leads to a current-locking phenomenon, when switching of one JJ drags several neighbors to the resistive state, thereby cascade-multiplying the readout voltage[46–48].

Here we demonstrate prototypes of cascade SCD based on a linear Nb-array and a stack of IJJs in a whisker-type Bi-2212 single crystal. We investigated MW and THz responses and compared single-junction and cascade SCD operation on the same device. The absorption efficiency is analyzed by studying the MW polarization loss diagrams. We discuss the operation principle and the ultimate performance of

cascade SCD. The advantage is shown to be associated with a new, pure cascade gain operating mode, allowing both cascade amplification of the sensitivity and reduction of statistical noise.

## Results

Cascade multiplication is widely used in photon detectors, such as photomultipliers and avalanche photodiodes[3]. Cascading increases the sensitivity $S$ (V/W) in proportion to the number of stages, $S_n = nS_1$. As sketched in Fig. 1a, this will reduce $NEP = \delta V/S$, provided the noise floor, $\delta V$ (V/Hz$^{1/2}$), is not multiplied equally.

We aim to achieve cascade multiplication in JJ arrays. Individual JJs in an array have inevitable variations. Without interjunction coupling, the low-power response is determined by the weakest single JJ. With

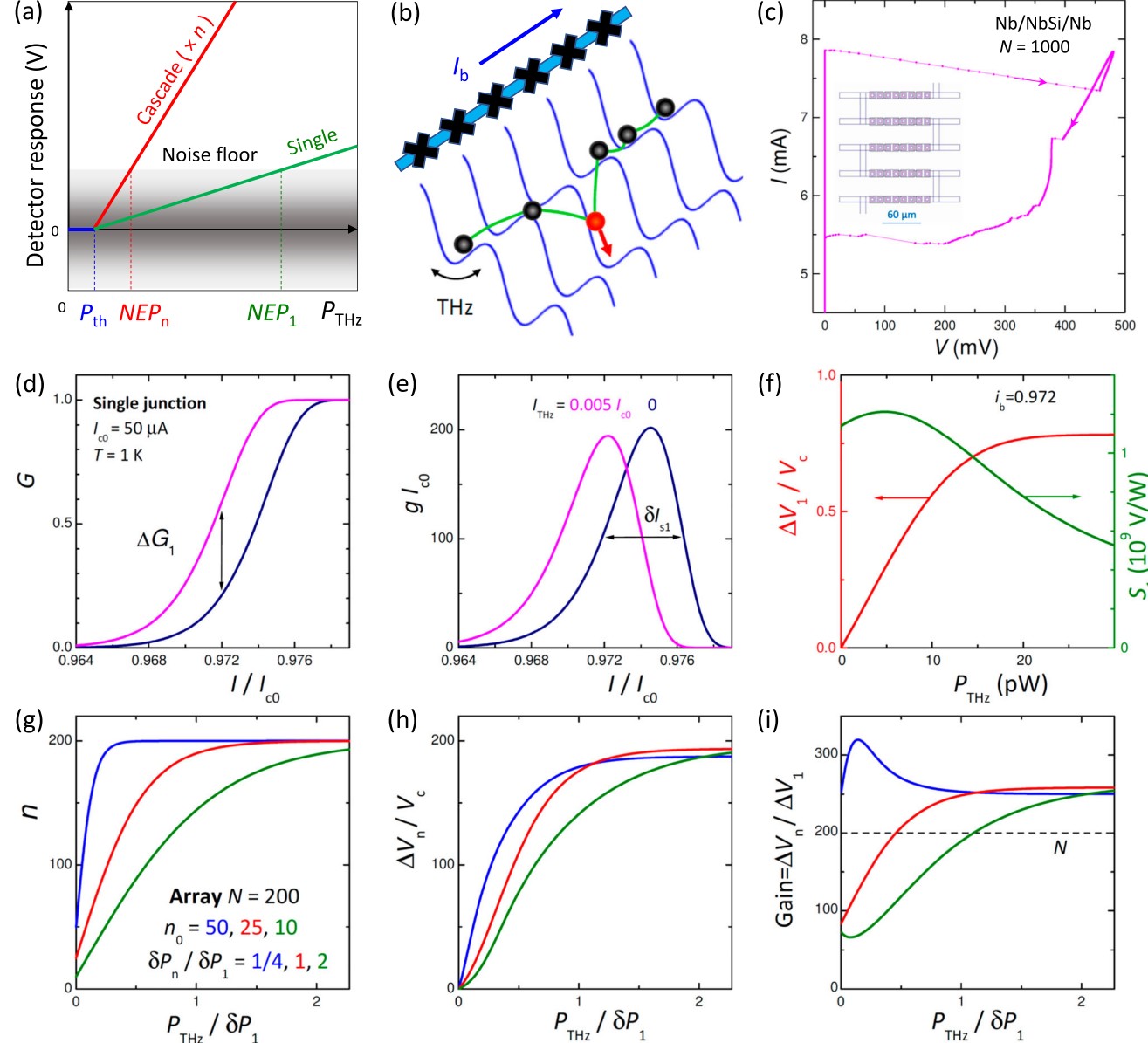

**Fig. 1 | Operation principle. a** A concept of cascade amplified detector. **b** The energy-phase diagram of a Josephson junction array. Switching (escape) of one junction leads to avalanche-like switching of neighbors due to the interjunction coupling. **c** The current-voltage characteristics of a linear array with $N = 1000$ Nb/Nb$_x$Si$_{1-x}$/Nb junctions at $T \simeq 2.5$ K. A nearly perfect current locking of all junctions can be seen. The inset shows the array layout. **d–f** Operation of a single-junction detector, SCD$_1$. Calculated switching probabilities **d** and probability densities **e** without (navy) and with (magenta) radiation. (f) Calculated voltage response

(red) and sensitivity (olive line) for $i_b = I_b/I_{c0} = 0.972$. Based on the data from ref. 15. **g,h** Operation of a cascade detector, SCD$_n$. **g** Three examples of active junction number distribution, $n(P_{THz})$, for arrays with $N = 200$ junctions: uniform ($\delta P_n = \delta P_1/4$, $n_0 = 50$, blue), intermediate ($\delta P_n = \delta P_1$, $n_0 = 25$, red), nonuniform case ($\delta P_n = 2\delta P_1$, $n_0 = 10$, olive), where $\delta P_1$ is the response range of a single junction, the red line in (**f**). **h** Array responses, calculated from Eq. (5). **i** Cascade gains for the three cases from (**g**). Note that they can be larger than $n$.

coupling, a current-locking phenomenon occurs[46], causing several JJs to switch simultaneously, as sketched in Fig. 1b. Figure 1c shows the current-voltage (I-V) characteristics of a linear array of $N = 1000$ Nb/Nb$_x$Si$_{1-x}$/Nb JJs[49]. It exhibits a nearly perfect current locking of all JJs, leading to a high readout voltage, $V \simeq 0.45$ V.

JJ arrays have been known to be advantageous for photon detection: arrays can help with impedance matching and broaden the dynamic range[50,51]. However, earlier studies focused on heterodyne mixers, operating in the quasiparticle branch of the I−V. In contrast, here we consider SCD, where the signal is generated upon switching out of the superconducting state.

## Operation principle

The JJ dynamics is equivalent to motion of a particle in a tilted washboard potential[26,34], Fig. 1b. The potential well is determined by the Josephson energy $E_{J0} = (\Phi_0 / 2\pi) I_{c0}$, where $\Phi_0$ is the flux quantum and $I_{c0}$ is the fluctuation-free critical current. The SCD is biased by an ac-current with frequency $f_b$ and amplitude $I_b \lesssim I_{c0}$. Transition to the resistive state occurs at a switching current $I_s < I_{c0}$. The premature escape from the well is caused by internal thermal or quantum fluctuations[15,26,34], and by the high-frequency current $I_{THz}$, induced by the incoming THz power[15],

$$P_{THz} = \frac{I_{THz}^2 R_{THz}}{2\chi}. \qquad (1)$$

Here $R_{THz}$ is the real part of the THz impedance.

Figure 1d−f summarize operation of an SCD$_1$, based on simulations from ref. 15 for an underdamped JJ with the quality factor $Q_0 = 100$, $I_{c0} = 50$ μA, $f_b = 150$ Hz, $V_c = 20$ mV (Bi-2212), $T = 1$ K, nonresonant escape, optimal $\chi = 0.5$ and $R_{THz} = 100$ $\Omega$[52]. For more details, see ref. 15 and Supplementary sections SI−III.

Figure 1d, e show switching probabilities, $G(I)$, and probability densities, $g = dG/dI$ (switching current histograms) without and with radiation. The response of SCD$_1$ is caused by the suppression of $I_s$, which shifts histograms and increases switching probability $\Delta G(I = I_b)$, leading to the mean voltage rise,

$$\Delta V_1 \simeq \Delta G(I_b) V_c. \qquad (2)$$

The red line in Fig. 1f shows the response at $I_b = 0.972 I_{c0}$. It saturates when $\Delta I_s$ exceeds the full-width at half-maximum (FWHM) of the switching histogram,

$$\delta I_{s1} \simeq 0.57 I_{c0} \left[ \frac{k_B T}{2 E_{J0}} \right]^{2/3}. \qquad (3)$$

The sensitivity, $S_1 = \Delta V_1 / P_{THz}$, shown by the olive line in Fig. 1f, is limited only by the finite $\delta I_{s1}$.

A switching current is reached twice per bias period at positive and negative current maxima. The probabilistic (binomial) switching leads to a telegraph noise[15],

$$\delta V_1 \simeq V_c \sqrt{\frac{G(1-G)}{f_b}}, \qquad (4)$$

which determines the noise floor and limits NEP$_1$ to ~1 pW/Hz$^{1/2}$ for the case of Fig. 1d−f. Note that NEP$_1$ does not depend on $V_c$ because it appears in both Eqs. (2) and (4). $V_c$ only affects the frequency range.

For a multi-junction cascade detector (SCD$_n$), the array response and sensitivity can be written as,

$$V_n \simeq G_n^* n V_c, \qquad (5)$$

$$S_n = V_c \left[ n \frac{\partial G_n^*}{\partial P} + G_n^* \frac{\partial n}{\partial P} \right], \qquad (6)$$

where $G_n^*$ is the probability of collective switching of $n$ JJs. The first term in Eq. (6) is caused by the shift of histograms, $\Delta I_s(P)$, similar to SCD$_1$. The second term represents a new detection mechanism specific for SCD$_n$, where the response $\propto \partial n / \partial P$, is caused by the pure cascade gain without shifting of $I_s$.

The switching statistics of an imperfect array are not binomial, but follow a compound distribution, where the output acquires a mean value, $n$, and a standard deviation $\delta n$. This changes the statistical noise,

$$\delta V_n = n V_c \sqrt{\frac{G_n^*(1 - G_n^*) + G_n^* \delta n^2 / 4 n^2}{2 f_b}}. \qquad (7)$$

In the ideal current locking case, $n = N$, $\delta n = 0$, Eq. (7) reduces to Eq. (4), but with a multiplied telegraph noise. This will not improve NEP. On the other hand, the uncertainty from the second term in Eq. (7) does not depend on $n$. Therefore, operation in the pure cascade gain mode, $G_n^* = 1$, allows the obviation of large telegraph noise, which improves NEP$_n$.

Figure 1g−i summarizes SCD$_n$ operation. The performance of SCD$_n$ depends on the distribution $n(P_{THz})$, with the most important parameters being the starting number, $n_0$, at $P_{THz} \to 0$ and the width $\delta P_n$.

Figure 1g shows three examples of $n(P_{THz})$ for arrays with $N = 200$ JJs. The leftmost (blue) is the case of a fairly uniform array with large $n_0 = 50$ and narrow $\delta P_n = 1/4 \, \delta P_1$; the middle (red) to $n_0 = 25$ and $\delta P_n = \delta P_1$; and the rightmost (olive) is to the least homogeneous array with small $n_0 = 10$ and broad $\delta P_n = 2 \, \delta P_1$. Here $\delta P_1$ is the response width of SCD$_1$, shown by the red line in Fig. 1f. Figure 1h represents the corresponding array responses, $\Delta V_n$. It is seen that a wider $\delta P_n$ expands the dynamic range but reduces the response.

Figure 1i shows the cascade gain $\Delta V_n / \Delta V_1$. Remarkably, it can be significantly larger than $n$. The overshooting is caused by the second term in Eq. (6), which provides the key advantage of SCD$_n$ and raises the signal-to-noise ratio by increasing the signal and obviating telegraph noise. Interestingly, this pure cascade gain term is associated with an imperfect current-locking, $\partial n / \partial P$, implying that some imperfection is beneficial for SCD$_n$.

## Microwave detection

We studied MW and THz responses of linear Nb/Nb$_x$Si$_{1-x}$/Nb JJ arrays and stacked IJJs in Bi-2212 whiskers. Fabrication and physical properties of Nb arrays were described in refs. 49,53−55. Details about the fabrication and characterization of Bi-2212 devices can be found in refs. 29,32,56. All presented measurements were performed at ambient magnetic field and $T \simeq 3.3$ K. See Methods and Supplementary sec. SVII−IX for more experimental details and additional data.

For MW detection, devices were irradiated quasi-optically by a linearly polarized source at $f \simeq 74.5$ GHz. MW attenuation, μ, and polarization angle, $\Theta$, were adjusted by grid polarizers. A Golay cell detector was used to monitor the incoming MW power.

Figure 2a shows the I−Vs of a Nb array with $N = 128$ JJs, measured upon minor variation of the bias current in the absence of MW. Multiple branches correspond to a different number, $n$, of active JJs. The interjunction coupling in these arrays is mediated by surface plasmons[55] and manifested by the appearance of collective resonant steps in the I−V's[49].

Figure 2b shows the MW response of the array. With increasing $P_{MW}$, progressively more JJs switch to the resistive state (red line) leading to an increase in array voltage (blue symbols). A correlation $V \propto n$ illustrates the cascade amplification phenomenon. At higher power, all $N = 128$ JJs switch, and the responsivity $dV/dP$ is greatly

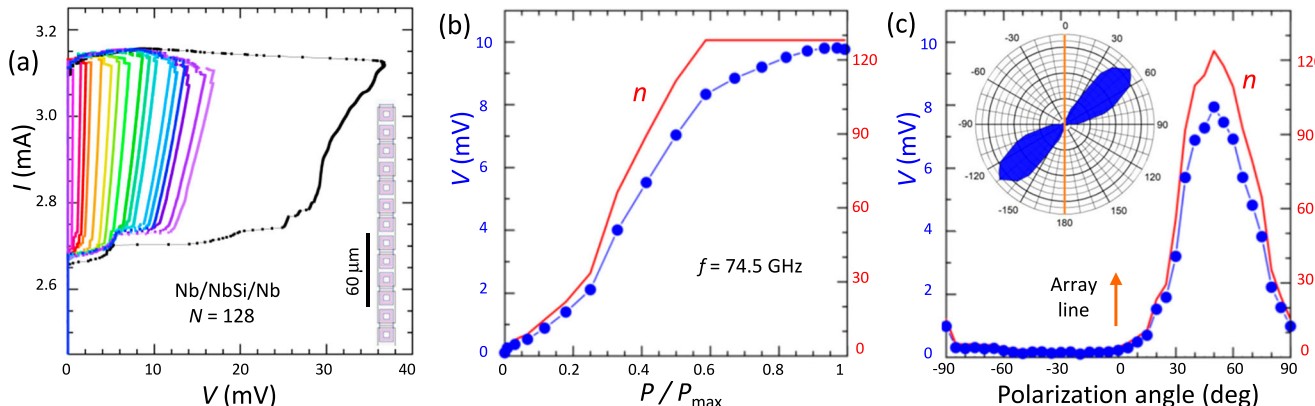

**Fig. 2 | Cascade detector based on a linear Nb/Nb$_x$Si$_{1-x}$/Nb array with N = 128 junctions. a** Ensemble of the I-V's without irradiation, obtained at slightly different bias currents. The inset shows the array layout. **b** The array voltage response (blue symbols, left axis) and the number of active junctions, $n$, (red line, right axis) as a function of microwave power, $f = 74.5$ GHz, normalized on $P_{max} \simeq 14$ nW. **c** Array response, $V$ and $n$, as a function of the polarization angle, $\Theta$, at constant microwave power. The inset shows the polarization loss diagram. The orange line marks the array direction. Profound off-axis lobes manifest the traveling-wave antenna effect.

reduced. In this case, it is due solely to the decrease of $I_s(P)$, the first term in Eq. (6), qualitatively similar to a single-JJ SCD$_1$[15,29].

To analyze the absorption efficiency, we measured the polarization loss diagram[57]. Figure 2c shows the array voltage (blue symbols) and the number of active JJs (red line) as a function of the polarization angle, $\Theta$, at a constant $P_{MW}$. $\Theta = 0$ corresponds to the MW electric field parallel to the array line. The inset displays the polarization-loss diagram, $\Delta V/V_G(\Theta)$, where $V_G \propto P_{MW}$ is the Golay cell voltage (constant in this experiment). The diagram exhibits two profound lobes at $\Theta \simeq 50°$ and $-130°$ with respect to the array. The off-axis behavior is consistent with the traveling-wave antenna operation of long electrodes reported for such arrays[54,55]. The estimated optical absorption efficiency of this array is $\chi \simeq 7\%$ (see Methods).

Figure 3a represents a scanning electron microscope (SEM) image of one of the Bi-2212 devices based on whisker-type single crystals[32]. Several mesas, containing stacked IJJs, were formed at the intersection of the whisker with the top gold electrodes, as sketched in Fig. 3f. To reduce $I_c$ and increase sensitivity, some mesas were trimmed using focused ion beam[29,56].

Figure 3b shows two I-V's of a mesa ($\sim 5 \times 5$ μm$^2$), with $N \simeq 250$ IJJs. The blue curve, with $n = 117$ active IJJs, was measured with $I_b$ slightly above the mean switching current, $I_s \sim 50$ μA. The multi-branch structure ends at $V \simeq 5$ V, corresponding to the large $V_c \simeq 20$ mV per IJJ. Switching of the first and the last IJJ occurs within an interval, $\delta I_n \sim 5$ μA $\sim 0.1 I_s$. Statistical analysis of multi-junction switching can be found in the Supplementary sec. SIV.

The inset in Fig. 3c shows a switching histogram (red) for a single IJJ in another mesa. The solid line is the expected probability density at the base $T$. Red circles in the main panel show the measured $T$-dependence of the FWHM. The blue line was obtained from Eq. (3). Overall agreement is excellent, indicating that IJJs are well described by the SCD formalism[58].

Figure 3d shows the I-Vs of the third mesa at different MW powers and $I_b = 38.1$ μA. The cascade gain depends on $I_b$. Figure 3e represents the MW power dependencies of $V$ (symbols) and $n$ (lines), measured at two biases. At lower $I_b = 25.3$ μA, the sensitivity, $dV/dP$, remains small up to some threshold, $P_{MW} \sim 0.2 P_{max}$. At higher $I_b = 38.1$ μA, there is no threshold and the sensitivity is high at $P_{MW} \to 0$. With a further increase of $I_b$, the sensitivity remains high at $P_{MW} \to 0$, but saturates at successively smaller power, thus reducing the dynamic range[15]. $I_b = 38.1$ μA is the optimal bias amplitude with both high sensitivity and large dynamic range. For comparison, in the Supplementary sec. SIX we analyzed the ordinary SCD operation on the same mesa using a single surface IJJ. The surface IJJ had a much

smaller $I_s$[38] and could be measured without activation of other IJJs[29]. The maximum single-junction response is almost 1000 times smaller than for the cascade SCD in Fig. 3e at the same MW power for the same mesa.

Figure 3f shows the polarization-loss diagram of this mesa. It has a four-fold shape with two smaller lobes aligned with the whisker. The larger lobes are approximately perpendicular, slightly inclined in the direction of one of the bias electrodes (see the Supplementary sec. SIX for more details). The large-scale electrode geometry of this device is shown in the background (yellow). The observed four-fold diagram is consistent with the turnstile-antenna geometry of the device[59,60], as sketched in the inset. The estimated optical absorption efficiency, $\chi \simeq 9\%$ (see Methods), is consistent with the emission efficiency reported for similar mesas[32].

## Detection of THz radiation from a Bi-2212 mesa

To verify THz operation, we performed an in-situ generation-detection experiment in which one mesa emits radiation and a nearby mesa detects it[29,32]. Figure 4a shows the I−V of a generator mesa ($\sim 10 \times 15$ μm$^2$), containing $N = 160 \pm 10$ IJJs. We analyzed the downturn part of it (magenta) when all IJJs are active. In this case, the Josephson frequency is well defined, $f_J = 2eV/hN$, as indicated by the top axis. The emission occurs at the cavity modes in the mesa[29,43,44]. The key signature of such emission (as opposed to heating) is the non-monotonous dependence of the signal on the dissipation power, $P_{gen} = I_{gen}V_{gen}$, with distinct peaks at cavity resonances[29].

Figure 4b shows low-bias part of the I−V of a detector mesa ($\sim 15 \times 15$ μm$^2$), about 22 μm away from the generator. A close-up in Fig. 4c represents the I−V of the weak surface junction. It had $I'_c \simeq 15$ μA, significantly lower than $I_c \simeq 370$ μA for the rest of the IJJs. This disparity allowed SCD$_1$ measurements at $I'_c < I_b \ll I_c$[29]. Cascade detection could be performed on the same mesa by increasing the bias to $I_b \sim I_c$.

Figure 4d shows SCD$_1$ (olive, $\times 50$) and SCD$_n$ (magenta) responses measured at $I_b = 25$ and 370 μA, respectively. It can be seen that (in this case) the resolution of SCD$_1$ is not sufficient to resolve the emission. The SCD$_n$ shows a much larger response with clear emission peaks. The main emission for this mesa occurs in the range from $\sim 4$ to 8.4 THz, as indicated by the top axis. The upper frequency is limited by self-heating of the generator mesa at large bias[32,42], which also leads to a monotonous upward drift of the detector signal at $P_{gen} > 2$ mW (see Supplementary sec. SX for the discussion of self-heating).

Figure 4e shows the detector I-V's measured with $I_b = 370$ μA at several bias points of the generator, marked by the same-color circles

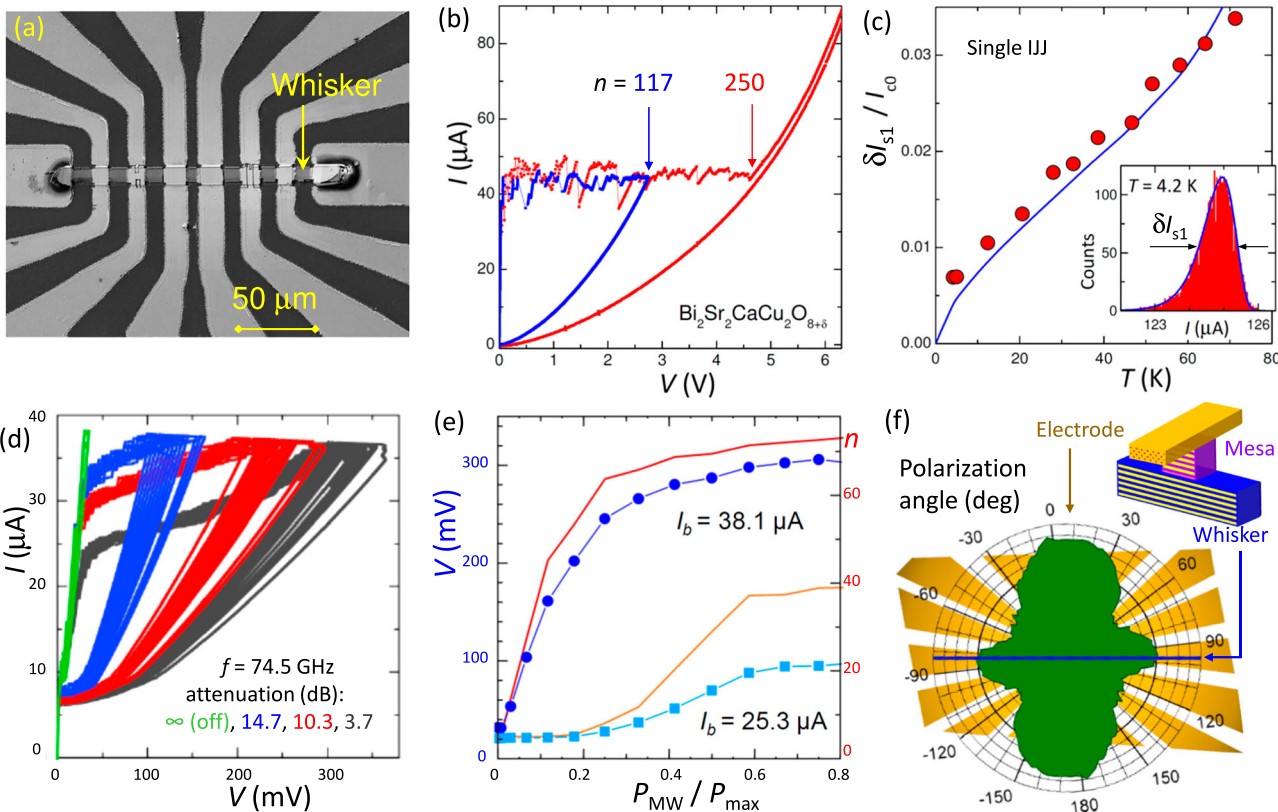

**Fig. 3 | Microwave characteristics of a Bi-2212 mesa. a** A scanning electron microscope image of a whisker-based device. **b** Two $I$-$V$ characteristics of a mesa with $N = 250$ intrinsic junctions, measured at different bias amplitudes in the absence of irradiation. **c** Red circles represent measured temperature dependence of the width of switching current histograms for a single intrinsic junction. The blue line is calculated from Eq. (3). The inset shows the measured switching current histogram (red) and the calculated switching probability density (blue line) at $T = 4.2$ K. **d** Ensemble of the $I$-$V$s for different microwave (MW) powers, determined by the attenuation factor μ, at constant $I_b = 38.1$ μA. **e** The mesa voltage (symbols) and the number of active junctions, $n$, (lines) versus the MW power, for $I_b = 25.3$ and 38.1 μA. **f** The polarization-loss diagram. The blue line indicates the orientation of the whisker. The yellow background shows the large-scale electrode geometry. The four-fold symmetry is consistent with the turnstile-antenna geometry of the device, as sketched in the inset.

in Fig. 4a, d. It can be seen that the cascade amplification factor, $n$, plays a dominating role in the detector response.

## Discussion

We observed similar behavior for low- and high-$T_c$ devices, either linear arrays or stacks. Both types of detectors have some advantages. 2D-linear arrays facilitate a large absorption area and simple scalability to tens of thousands of JJs[53], while Bi-2212 devices can operate in the whole THz range [0.1-10] THz[29].

MW measurements, performed under identical conditions, allow for quantitative comparison of the two devices (see Methods). Although the sensitivity of Bi-2212 is almost hundred times higher, the NEP's are similar $\simeq 3$ pW/Hz$^{1/2}$. This is a consequence of dominant telegraph noise, Eq. (4), which scales with $V_c$, making the NEP approximately independent of $V_c$. Therefore, the relevant SCD figure of merit is NEP$_n$, rather than $S_n$.

The new detection mechanism, when the response is caused by the pure cascade gain without suppression of $I_s$, is central for SCD$_n$. It increases sensitivity, leading to overshooting, $V_n/V_1 > n$. For example, for the olive line in Fig. 1i, the gain at $P_{THz} \rightarrow 0$ is 7.3 times greater than $n_0 = 10$. Even more importantly, this mechanism allows for partial obviation of the telegraph noise.

The statistical uncertainty of an array, Eq. (7), is contributed by the telegraph noise (first) and the gain uncertainty (second term). Unlike SCD$_1$, SCD$_n$ can operate at $G_n^* = 1$, when the array always switches into the resistive state at every bias cycle and the signal is carried only by the pure cascade gain, $\partial n/\partial P$. In this case the telegraph noise term

vanishes, the noise reduces to

$$\delta V_n = V_c \frac{\delta n}{2\sqrt{2f_b}}, \quad (8)$$

the sensitivity is $S_n = V_c \partial n/\partial P \simeq V_c N/\chi \delta P_n$, yielding

$$NEP_n \simeq \frac{\delta n \delta P_n}{2\chi N\sqrt{2f_b}}. \quad (9)$$

The $N^{-1}$ dependence reflects the benefit of cascading.

All measurements presented here were performed in the pure gain mode, with the exception of high power range when suppression of $I_s$ also contributed to the response. The latter is clearly distinguishable in Fig. 2b at $P/P_{max} > 0.6$, when cascade gain is saturated, $n = N$. Apparently, the ordinary sensitivity in this range is much smaller than the cascade sensitivity at lower power.

The MW NEPs of both detectors, ~3 pW/Hz$^{1/2}$, are consistent with Eq. (9). They are modest due to low $f_b = 23$ Hz and imperfect impedance matching, $\chi < 0.1$. Similar arrays ($T = 3$ K, $I_{c0} = 50$ μA) with slightly more JJs, optimal $\chi = 0.5$ and higher $f_b = 230$ kHz would reach $NEP$ ~ 1 fW/Hz$^{1/2}$.

The ultimate limit of NEP is determined by the two main tuning parameters of SCD: $T$ and $I_{c0}$.

The decrease of $T$ leads to freezing out of thermal fluctuations and shrinking of switching histograms, Eq. (3). This enhances the sensitivity $\propto 1/\delta I_{s1}$ (see Supplementary sec. III). The limit is set by the

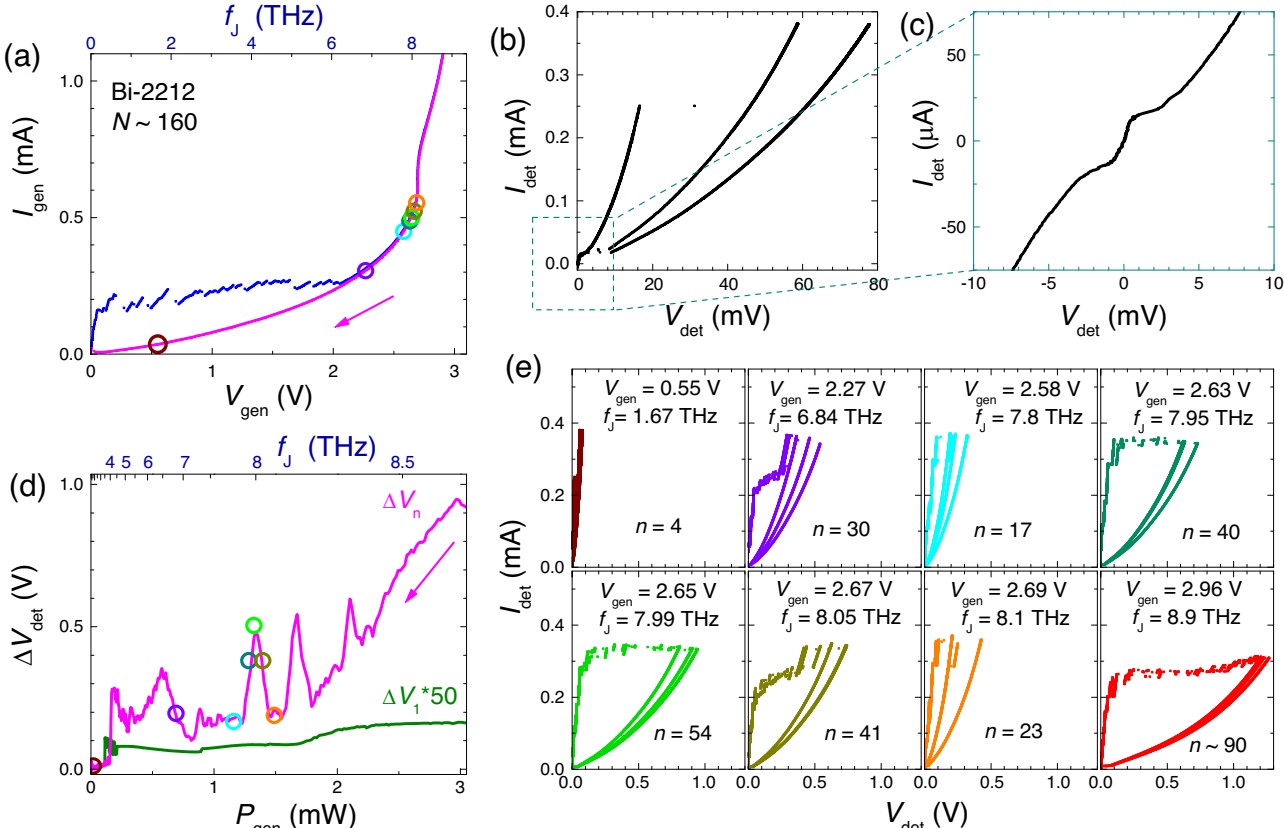

**Fig. 4 | Detection of terahertz (THz) radiation from a Bi-2212 mesa. a** The $I-V$ characteristics of the generator mesa for upward (blue) and downward (magenta) bias sweeps. Top axis indicates the anticipated Josephson frequency. **b, c** Low-bias $I-V$'s of the detector mesa with **c** showing the $I-V$ of the weak surface junction. **d** Measured detector responses versus total generator power, $P_{gen} = I_{gen}V_{gen}$. The magenta curve represents the cascade detector mesa response. The sharp peaks correspond to emission at cavity mode resonances in the generator mesa, the monotonous background is caused by self-heating. The olive line represents a single surface junction response (multiplied by a factor 50) on the same mesa. **e** A set of detector $I-V$ characteristics with fixed $I_b = 370\,\mu A$ at different bias points of the generator, marked by the same-color circles in (**a**) and (**d**). It can be seen that the response is caused predominantly by the cascade gain.

crossover temperature to macroscopic quantum tunneling (MQT)[4,22,23,26,28,34,58], qualifying SCD as a quantum-limited detector. For $T_{MQT} = 30\,mK$ the improvement will be a factor $[3.3K/30mK]^{2/3} \simeq 23$, bringing NEP at $I_{c0} = 50\,\mu A$ to the ~ 50 aW/Hz$^{1/2}$ level.

The decrease of $I_{c0}$ lowers $E_{J0}$ and increases sensitivity approximately as $S_1 \propto 1/I_{c0}$[15]. However, at $I_{c0} \lesssim k_B T/\Phi_0$ the potential well becomes so low that thermal fluctuations become capable of moving the particle even without tilt/current. The JJ then enters the phase-diffusion state[35], which sets the limit of SCD sensitivity[15]. However, phase diffusion is not necessarily detrimental. In this state, the JJ is switching in and out at a frequency close to the Josephson plasma frequency $f_p$ ~ 100 GHz. Paradoxically, such chaotic dynamics could effectively average away the telegraph noise[15]. Evidence for the collapse of visible fluctuations at the edge of phase diffusion has been reported[58].

Thus, we anticipate that the ultimate NEP is achieved at $T = T_{MQT}$ and $I_{c0}^* \sim k_B T_{MQT}/\Phi_0$. For $T_{MQT} = 30\,mK$, $I_{c0}^* \sim 6\,nA$ is almost four orders of magnitude smaller than $50\,\mu A$ assumed in estimations above. This indicates potential for a drastic reduction of NEP, provided the impedance matching and absorption efficiency could be preserved. The ultimate limit of NEP in such quantum phase diffusion state remains an interesting question for further investigation.

In conclusion, we have introduced the concept of a cascade SCD based on arrays of coupled JJs and demonstrated prototypes utilizing both low- and high-$T_c$ superconductors. Counterintuitively, some inhomogeneity has been shown to be beneficial for cascade SCD

because it enables a new operation mode, which enhances the detector sensitivity while simultaneously reducing statistical noise, thus lowering the noise equivalent power. We developed qualitative and quantitative methods to evaluate antenna efficiency and optical absorption and showed that Bi-2212 detectors operate effectively across a broad THz frequency range.

## Methods
### Experimental details
Measurements were performed in a closed-cycle optical cryostat with a base temperature ~3.3 K at zero magnetic field. Arrays were current-biased using a programmable source. The $I-V$s were measured in a quasi 4-probe configuration. Additional information about experimental setup, bias configuration, and microwave measurements can be found in sec. SVII of the Supplementary. Details about fabrication and characterization of Nb arrays and Bi-2212 mesas are provided in sec. SVIII and SIX.

The operation of SCD requires statistical analysis. The switching statistics of Bi-2212 mesas for single and multi-junction switching can be found in sec. SIV of the Supplementary. It is well described by conventional thermal-activation theory, described in sec. SII.

The SCD response was measured via lock-in measurements. For this, a harmonic current, $I = I_b \sin(2\pi f_b t)$, with the amplitude $I_b$ and frequency $f_b = 23\,Hz$ was supplied for a time interval of 1 s. The read-out voltage corresponds to the Fourier component of the detector $V(t)$ wave form at $f = f_b$. The lock-in response was calculated on-flight by the

FPGA-based FFT routine. Additional information about SCD operation and lock-in readout can be found in sec. SI and SV.

## Quantitative estimation of detector characteristics

The incoming MW power was measured using a cryostat as a bolometer. The sample stage in our cryostat has a thermal resistance, $R_{th} \simeq 0.5$ K/mW[32]. By measuring a small temperature rise $\Delta T \sim$ mK, caused by the MW beam, we directly measured the total incoming power, $P_{tot} = \Delta T/R_{th}$. Due to the large $\lambda_0 \simeq 4$ mm and significant diffraction at several narrow ($\simeq 1$ cm) apertures of the cryostat, the MW power density was fairly uniform, $q = P_{tot}/\pi r^2$. Here, $r \simeq 1.5$ cm is the radius of the sample space. MW power impacting on SCD was calculated as $P = qA$, where $A \simeq 1$ mm$^2$ is the effective absorption area of the receiving antenna, formed by the electrodes. Thus, we obtained $P_{max} \simeq 14$ nW.

The calibration of $P_{MW}$ allows a straightforward estimation of the linear optical response at low power, using the $V(P)$ dependencies from Figs. 2b and 3e. For Bi-2212, $S_{MW} \simeq 1.3\,10^8$ V/W. For the Nb-array, $\partial V/\partial P$ is changing with $P$, Fig. 2b. The variation is caused by the one-polarizer configuration used in this measurement, in combination with the sharp polarization diagram, Fig. 2c. The sensitivity at the lobe is $\sim 2\,10^6$ V/W.

The noise measurements presented in Supplementary Fig. S6 reveal a statistical uncertainty, $\delta V \simeq 2\,10^{-4}$ V/Hz$^{1/2}$ for the Bi-2212 device. This is consistent with Eq. (8) for the actual $V_c \simeq 5$ mV, $f_b = 23$ Hz, and $\delta n = 1$. The large uncertainty limits the optical MW NEP(Bi-2212) to ~3 pW/Hz$^{1/2}$.

Alternatively, the NEP can be estimated from Eq. (9). For the Bi-2212 detector, $V(P)$ in Fig. 3e indicates that $\delta P_n \simeq 0.2 P_{max}$. For the actual parameters ($\delta n = 1$, $N = 74$), this results in essentially the same optical $NEP_{MW} \simeq 2.8$ pW/Hz$^{1/2}$ for Bi-2212. Similar analysis for the Nb-array ($\delta n = 1$, $\delta P_n = 0.4 P_{max}$ see Fig. 2b, $N = 128$) yields $NEP_{MW} \simeq 3.2$ pW/Hz$^{1/2}$. The agreement in obtained NEP values confirms the validity of Eqs. (8) and (9).

At high power, when the suppression of switching current, $\Delta I_s$, becomes distinct, it is possible to directly estimate the absorbed power[29]:

$$P_a \simeq \frac{2\sqrt{2}}{3\pi}\left(\frac{\Delta I_s}{I_{c0}}\right)^{3/2} I_{c0} V_c. \tag{10}$$

For the Bi-2212 detector from Fig. 5d, $\Delta I_s \simeq 3$ μA at 3.7 dB attenuation. Taking $I_{c0} = 38$ μA, $V_c = 5$ mV we obtain $P_a \simeq 1.3$ nW. Thus, the optical MW absorption efficiency is $\chi = P_a/P_{MW} \simeq 9\%$. This value is comparable to the observed emission efficiency for similar Bi-2212 mesas (albeit at different frequencies)[32], supporting our calibration of $P_{max}$.

For the Nb array, a similar estimation yields $\chi \simeq 7\%$. The absorption efficiencies for both types of detectors are several times smaller than the optimal value, which for JJs is less than 50% due to the presence of the quasiparticle leakage current[24]. Thus, the impedance matching in our detectors is not ideal. Nevertheless, antenna elements on our devices provide decent coupling to the radiation field, as evidenced by the geometry-specific polarization loss diagrams.

For the THz generation-detection experiment in Fig. 4, we could not confidently estimate the incoming power. However, the absorbed power can be estimated in the same way using Eq. (10). For the green $I$-$V$ in Fig. 6e at $f_J = 7.99$ THz, the suppression of the switching current is $\Delta I_s = 35$ μA. With $I_{c0} = 390$ μA, $V_c = 16$ mV and $V \simeq 1$ V it yields $P_a \simeq 50$ nW and $S_a \simeq 2\,10^7$ (V/W). A similar analysis for the violet curve at $f_J = 6.84$ THz yields $S_a \simeq 3.5\,10^7$ V/W.

Note that the sensitivities obtained with the help of Eq. (10) are several times smaller than the estimated linear responses. The main reason for that is that a confident estimate of $\Delta I_s$ could be made only at high power, when detectors are already in the saturation state.

## Data availability

Data supporting the findings of this study are available from the manuscript and its Supplementary Information. The data is available from the corresponding author upon request.

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

## Acknowledgements

We are grateful to R. Gerdau and M. Galin for assistance with fabrication of Nb-arrays and to A. Kalenyuk for assistance with fabrication of Bi-2212 samples.

## Author contributions

R.G., K.I.S. and O.K. fabricated samples. R.G. and A.E.E. performed measurements. V.M.K. conceived the project and wrote the paper with input from all co-authors.

## Funding

## Competing interests

The authors declare no competing interests.
