## [Transparent Peer Review file · Nature Communications]

Cascade switching current detectors based on arrays of Josephson junctions

Corresponding Author: Professor Vladimir Krasnov

Version 0:

Reviewer comments:

Reviewer #1

(Remarks to the Author)

This manuscript proposes the use of a high-sensitivity terahertz detector by considering the cascade switching phenomenon induced by electromagnetic wave irradiation to intrinsic Josephson junction stacks of a high-temperature superconductor as cascade amplification. Terahertz waves are attracting attention as the next-generation communication technology, and high-temperature superconductors are expected to be a materials that can develop devices with characteristics that cannot be achieved with semiconductors due to their low dissipation and nonlinearity in the terahertz region. In this sense, the results described in this paper are extremely important not only in the field of superconductivity but also in photonics, and we judge that they are worthy of publication in Nature Communications. The argument is clear, it is very easy to read, and the presentation of the figures sufficiently supports our understandings, and the Supplementary information provides sufficient information to understand and reproduce the experimental data and analysis.

If I may make one comment, it is the detection of terahertz radiation from the IJJ generator shown in Fig. 6 (d). I agree that the peaky responses in ΔV_n can be considered as radiations from the IJJ generator, but there is a need to mention and analyse the gradual rise in the background. This is probably a response due to the temperature rise caused by the Joule heat generated in the device, but the fact that it is higher than the peaks in the region of low P_{gen} is confusing to the reader. It should be mentioned in the figure caption and discussed.

As a reviewer, I have a question: the response of the Joule heat response of the ΔV_n is expected to be linear with respect to P_{gen} , but it does not seem to be the case. The authors' opinion on this should be replied. By responding to this point, the response of ΔV_n will be more contrasted with the response of ΔV_{-1} , and it is expected that the cascade amplification effect will be shown more clearly.

Reviewer #2

(Remarks to the Author)

The manuscript tried to propose a novel cascade-amplified switching current detector (SCD) based on arrays of Josephson junctions (JJs). But many concepts easy lead the reader's confusions and should be made more clear before to be considered for the publication.

1) For the detectors, most important parameter is the noise equivalent power (NEP), which is related the sensitivity and noise performance. The manuscript given a result that the noise of detector is determined by the readout amplifier at room temperature. That means the real noise performance of the detector was not characterized and only the pre-amplifier was characterized. By the way, taking the result in the abstract with sensitivity better than 10^{13} V/W and the noise voltage of 1 nV/Hz^{1/2} for the amplifier, you can get the NEP of better than 10^{-22} W/Hz^{1/2} working at 3 K (?) and with broadband frequency band (from microwave to terahertz?). Is it possible? Please also consider the level of the back-ground noise at working temperature and 1/f noise of the amplifiers.

2) Also, the sensitivity of the detector is strong related the impedance matching, not only the output voltage. When the numbers of the JJs is increased, the impedance of the detectors will increase also. The mismatching between the device and antenna/free space as well as readout system will be changed very much. So the concept about the impedance matching should be made clear also. The manuscript should give the details about the impedance matching: what are the impedances of the device, antenna and readout system, how much of them...

3) About the microwave (MW) frequency range and terahertz (THz) frequency range, the manuscript mentioned the THz range of 0.1-10 THz. But the 74.5 GHz (0.0745 THz) was used for the sample at MW range. How about the performance difference between the frequencies at 0.0745 THz (MW band) and 0.1 THz (THz band)? Also, how about the performance

difference between the frequencies at 0.1 THz and 10 THz (same THz band)? Also, the impedances of devices at MW and THz should be defined and compared quantitatively...

4) About the single photon detection (SPD) from MW to THz band, it is better to make clear for how much of the performance (NEP) of the detectors can be satisfied the condition of the SPD at MW and THz band.

5) About the cascade, it should be made clear for the IJJs with a surface JJ of small critical current. When the larger bias current was added to such IJJs, it is already in a voltage state. How about the cascade in this case?

6) Another important thing is about the heating effect of the arrays, especially for the IJJs. When the devices work at the low temperature, the effective temperature will be changed very much when the voltage is appeared with EMW irradiations as well as the larger bias...

7) As the manuscript reminded the antenna effect is quite important for the detectors, it is suggested to design an effective antenna and research it in details and quantitatively.

8) Finally, as the optical NEP is not so difficulty for the experiments, such as that for the TED and KID with reasonable optical NEP. It is suggested to give the optical NEP for the detectors in the manuscript. Also, one of the best device working at one of best operating temperature and one of the best frequency with a best readout system should be good enough to demonstrate a new propose.

Version 1:

Reviewer comments:

Reviewer #1

(Remarks to the Author)

The authors gave satisfactory replies. However, the added explanation of detector response in Fig. 6(d) still requires quantitative descriptions: what causes higher ΔV_n in no-radiating $P_{gen} > 2.5$ mW than peaky responses attributed to THz radiations.

Reviewer #2

(Remarks to the Author)

Some of improvements in the revised manuscript have been made. But there is a big problem about estimated value of NEP. It is very difficult to get the NEP about $\text{zW/Hz}^{1/2}$ at THz band in practical. To avoid the misleading to the researchers, it is better to check the estimations again in details by the authors. For example, if you use the JJ with I_c of 10 nA and V_c of 20 mV (the values appeared in the paper), the R_n will be very larger even for single JJ, much larger will be happened for the array. In this case, you can not get any couple efficiency between the array and antenna/free space (larger impedance mismatching). Also, the heating and noise problems will happen in practical. These important facts should be checked and calculated in details.

Version 2:

Reviewer comments:

Reviewer #1

(Remarks to the Author)

The authors added satisfactory quantitative descriptions on detector responses with and without coherent terahertz radiations with its details being referred to the supplementary information. Therefore, I recommend to publish the manuscript with the present contents.

Reviewer #2

(Remarks to the Author)

As the best expected (Calculated) value of NEP has been removed, I do have no other more comments to the manuscript. The ideas using cascade one may be useful to the researchers.

Reply to Reviewer #1:

Reviewer #1 writes:

“This manuscript proposes the use of a high-sensitivity terahertz detector by considering the cascade switching phenomenon induced by electromagnetic wave irradiation to intrinsic Josephson junction stacks of a high-temperature superconductor as cascade amplification. Terahertz waves are attracting attention as the next-generation communication technology, and high-temperature superconductors are expected to be a materials that can develop devices with characteristics that cannot be achieved with semiconductors due to their low dissipation and nonlinearity in the terahertz region. In this sense, the results described in this paper are extremely important not only in the field of superconductivity but also in photonics, and we judge that they are worthy of publication in Nature Communications. The argument is clear, it is very easy to read, and the presentation of the figures sufficiently supports our understandings, and the Supplementary information provides sufficient information to understand and reproduce the experimental data and analysis.”

Reply-1:

Thank you for the encouraging comments. In the modified version we have introduced clarifications and modifications following the Reviewers critics.

Reviewer#1 writes:

“If I may make one comment, it is the detection of terahertz radiation from the IJJ generator shown in Fig. 6 (d). I agree that the peaky responses in ΔV_n can be considered as radiations from the IJJ generator, but there is a need to mention and analyse the gradual rise in the background. This is probably a response due to the temperature rise caused by the Joule heat generated in the device, but the fact that it is higher than the peaks in the region of low $P_{\{gen\}}$ is confusing to the reader. It should be mentioned in the figure caption and discussed.”

Reply-2:

Thank you for the suggestion. This is indeed the case. Following Reviewer’s suggestion, we added an additional statement in the figure caption.

Reviewer#1 writes:

“As a reviewer, I have a question: the response of the Joule heat response of the ΔV_n is expected to be linear with respect to $P_{\{gen\}}$, but it does not seem to be the case. The authors' opinion on this should be replied. By responding to this point, the response of ΔV_n will be more contrasted with the response of ΔV_1 , and it is expected that the cascade amplification effect will be shown more clearly.”

Reply-3:

Responses of single and cascade SCDs are illustrated in Figs. 1 (f) and (h). It can be seen that while $V_1(P)$ is approximately linear at low P , $V_n(P)$ is nonlinear. The nonlinearity arises from the power-dependence of the cascade amplification factor, $n(P)$, as shown in Fig. 1 (g). To address this issue, we added a new Fig. 1 (i) which shows the (nonlinear) ratio of voltage responses SCD_n/SCD_1 and a corresponding clarification/discussion in the text.

Reply to Reviewer #2

Reviewer #2 writes:

“The manuscript tried to propose a novel cascade-amplified switching current detector (SCD) based on arrays of Josephson junctions (JJs). But many concepts easy lead the reader’s confusions and should be made more clear before to be considered for the publication.

1) For the detectors, most important parameter is the noise equivalent power (NEP), which is related the sensitivity and noise performance. The manuscript given a result that the noise of detector is determined by the readout amplifier at room temperature. That means the real noise performance of the detector was not characterized and only the pre-amplifier was characterized. By the way, taking the result in the abstract with sensitivity better than 10^{13} V/W and the noise voltage of $1 \text{ nV/Hz}^{1/2}$ for the amplifier, you can get the NEP of better than $10^{-22} \text{ W/Hz}^{1/2}$ working at 3 K (?) and with broadband frequency band (from microwave to terahertz?). Is it possible? Please also consider the level of the back-ground noise at working temperature and $1/f$ noise of the amplifiers.”

Reply-1:

We are grateful to the Reviewer for raising this question. We admit that NEP analysis was superficial and estimation was confusing and incorrect. In the modified version we introduced extensive changes that address and correct NEP evaluation.

The major limiting factor of SCD NEP is the large statistical uncertainty (telegraph noise), caused by the probabilistic switching. Even though we have discussed it in the previous version, we managed to convince ourselves that it is possible to obviate this type of noise by a smart Fourier analysis. In reality, the smart analysis only reveals, but not removes it. The achieved NEP has been reevaluated drastically to $\sim \text{pW/Hz}^{1/2}$ level. Simultaneously we added a serious discussion of ultimate limitations in Discussions.

Reviewer #2 writes:

“2) Also, the sensitivity of the detector is strong related the impedance matching, not only the output voltage. When the numbers of the JJs is increased, the impedance of the detectors will increase also. The mismatching between the device and antenna/free space as well as readout system will be changed very much. So the concept about the impedance matching should be made clear also. The manuscript should give the details about the impedance matching: what are the impedances of the device, antenna and readout system, how much of them...”

Reply-2:

As we emphasized in the manuscript, the optimization of absorption efficiency requires implementation of an antenna, properly matched to a detector. In the modified version we added a quantitative estimation of the optical MW absorption efficiency (in Methods) plus statements in the text. For the Bi-2212 detector it is $\chi=9\%$, consistent with the reported emission efficiency from similar mesas [32]. For Nb arrays, $\chi=7\%$.

The absorption efficiencies for both types of detectors are several times smaller than the optimal value, which for JJs is less than 50% due to the presence of the quasiparticle leakage current [24]. Thus, the impedance matching in our detectors is not ideal. Nevertheless, antenna elements on our devices provide decent coupling to the radiation field, as evidenced by the geometry-specific polarization loss diagrams.

Concerning the impedance mismatch: this issue has been expanded in the new paragraph #3 of the introduction, where we explained that the main origin of mismatch is the size mismatch

of the large wavelength and the small detector, which requires implementation of antenna elements with the size $\sim\lambda_0$. Linear arrays enable good coupling in a straightforward way by increasing the array size and absorption area. For compact Bi-2212 stacks the specific antenna elements are required. A patch antenna design has been proposed in [24,44]. Here we exploit a turnstile antenna geometry [59]. Generally, the implementation of arrays does help to improve impedance matching, as it has already been shown earlier [50,51], because this adds a flexibility in the impedance matching problem in terms of the array geometry and number of junctions.

Reviewer #2 writes:

“3) About the microwave (MW) frequency range and terahertz (THz) frequency range, the manuscript mentioned the THz range of 0.1-10 THz. But the 74.5 GHz (0.0745 THz) was used for the sample at MW range. How about the performance difference between the frequencies at 0.0745 THz (MW band) and 0.1 THz (THz band)? Also, how about the performance difference between the frequencies at 0.1 THz and 10 THz (same THz band)? Also, the impedances of devices at MW and THz should be defined and compared quantitatively...”

Reply-3:

The frequency limit is set by the characteristic voltage V_c and the Josephson plasma frequency.

For Nb-arrays from the same batch, the upper experimentally confirmed frequency is ~ 0.5 THz [Galín et al., Linewidth Measurements of a Large Niobium Josephson Junction Array, IEEE Trans. Appl. Supercond. 34, 1100405 (2024)]. The difference in operation at 0.075 THz compared with 0.1 THz is negligible. Therefore, our results remain valid.

Bi-2212 mesas operate in the whole THz range [29]. However, the detector response in such a broad range does depend on frequency, not the least due to the response function of the matching antenna, which selects (limits) the specific frequency band. Interestingly, the quantitative estimation of MW absorption efficiency (9%) seems to be compatible with the THz emission efficiency (up to 12%). The statement is added in the modified version.

Reviewer #2 writes:

“4) About the single photon detection (SPD) from MW to THz band, it is better to make clear for how much of the performance (NEP) of the detectors can be satisfied the condition of the SPD at MW and THz band.”

Reply-4:

This refers to a sentence in the Introduction, which summarizes statements from previous works. We agree that it is very short and cryptic. Note that we intentionally used the word “potential” to soften the statement because to our knowledge there has been no convincing SPD demonstration. Some researchers call photon number sensitivity in circuit QED (which essentially use SCD) as SPD, but this is a stretching of the SPD concept. As discussed in [15], SCD is not well suited for SPD because it has a 50% dark count rate at the optimal bias amplitude. To reach SPD, the dark count rate has to be drastically reduced by reducing the bias current amplitude, which simultaneously reduce the sensitivity. To avoid the confusion, we removed this statement.

Reviewer #2 writes:

“5) About the cascade, it should be made clear for the IJJs with a surface JJ of small critical current. When the larger bias current was added to such IJJs, it is already in a voltage state. How about the cascade in this case?”

Reply-5.

The weak JJ creates a small offset voltage, V_0 , at $P=0$, which is independent of the number of switched stronger JJs and does not add the statistical uncertainty (because the switching probability $G=1$). Note that the voltage offset is always present in SCD, both single junction and cascade. The detector response is defined as $\Delta V = V(P) - V_0$. Therefore, this is not a problem. We added corresponding clarifications in the last paragraph of the Supplementary Sec. SV.

Reviewer #2 writes:

“6) Another important thing is about the heating effect of the arrays, especially for the IJJs. When the devices work at the low temperature, the effective temperature will be changed very much when the voltage is appeared with EMW irradiations as well as the larger bias...”

Reply-6:

Indeed, heating of the detector by the dissipation in the generator mesa leads to a systematic drift of the detector characteristics. This leads to a monotonous upturn of the detector signal in Fig. 4(d) at $P_{gen} > 2\text{mW}$. The key difference between heating and EMW emission is that heating response is monotonous with respect to P_{gen} , while the emission occurs at specific cavity-mode frequencies and is non-monotonous vs. P_{gen} . Additional clarifications are provided in the modified version.

Reviewer #2 writes:

“7) As the manuscript reminded the antenna effect is quite important for the detectors, it is suggested to design an effective antenna and research it in details and quantitatively.

Reply-7:

The analysis of antenna elements has already been presented earlier. For large Nb arrays, numerical analysis of active travelling wave antenna performance has been analyzed in Ref. [54] and in [Supercond. Sci. Technol. **34** 075005 (2021)]. For Bi-2212 mesas on whiskers the turnstile antenna simulations at THz frequency were presented in [59]. We added this reference in the modified version. The presented polarization-loss diagrams provide a qualitative experimental confirmation of the antenna efficiency. This is new and important. We note that although the reported values of optical MW absorption efficiency 7-9% are less than 50%, however the optimal value in JJs is always reduced ($\chi < 50\%$) by the quasiparticle leakage current, as discussed in [24]. This is particularly relevant for Nb/NbSi/Nb which are not strongly underdamped. A corresponding statement is added.

Reviewer #2 writes:

“8) Finally, as the optical NEP is not so difficulty for the experiments, such as that for the TED and KID with reasonable optical NEP. It is suggested to give the optical NEP for the detectors in the manuscript. Also, one of the best device working at one of best operating temperature and one of the best frequency with a best readout system should be good enough to demonstrate a new propose.”

Reply-8:

The MW optical sensitivity are described in sec. “Quantitative estimation of detector characteristics” of Methods. In the modified version we specified that the characteristics are “optical”. In this work we primary aim to present the new concept of cascade SCD, rather than to reach the ultimate performance. However, in the modified version we do provide a correct estimation of the ultimate limit and ways to reach it.

Finally, we want to thank the Reviewer for the in-depth critical analysis of the manuscript that revealed an important shortcoming. In the modified version we introduce significant changes correcting the mistake, and providing an in-depth clarification of SCDn performance.

Reply to Reviewers

Reviewer #1

Reviewer #1 writes:

“The authors gave satisfactory replies. However, the added explanation of detector response in Fig. 6(d) still requires quantitative descriptions: what causes higher ΔV_n in no-radiating $P_{gen} > 2.5$ mW than peaky responses attributed to THz radiations.”

Reply:

To answer these questions, we provided a quantitative analysis and extensive discussion of self-heating in the new sec. SX of the Supplementary. We also added a short note with the reference to the Supplementary in the main text.

Reviewer #2 (Remarks to the Author):

Reviewer #2 writes:

“Some of improvements in the revised manuscript have been made. But there is a big problem about estimated value of NEP. It is very difficult to get the NEP about $\mu\text{W}/\text{Hz}^{1/2}$ at THz band in practical. To avoid the misleading to the researchers, it is better to check the estimations again in details by the authors. For example, if you use the JJ with I_c of 10 nA and V_c of 20 mV (the values appeared in the paper), the R_n will be very larger even for single JJ, much larger will be happened for the array. In this case, you can not get any couple efficiency between the array and antenna/free space (larger impedance mismatching). Also, the heating and noise problems will happen in practical. These important facts should be checked and calculated in details.”

Reply:

The Reviewer correctly noted that R_n for $I_c=6\text{nA}$ will be very large (in the M Ω range). However, just R_n is not a key factor for impedance matching. The ideal SIS tunnel junction at $T=0$ has an infinite sub-gap quasiparticle resistance, but, as emphasized in Red. [24], this is actually good because it removes the leakage current and thus increases the maximum absorption efficiency (to 0.5). What matters is the real part of antenna impedance. It is determined by the total quality factor, Q_{eff} , which depends on all sources of THz dissipation, most noticeably from surface resistance, dielectric losses, and in JJs the quasiparticle leakage current: $1/Q_{eff}=1/Q_{qp}+1/Q_{surf}+1/Q_{diel}+\dots$ (Eq. (45) in Ref. [24]). When $R_{qp}=\infty$, the first term vanishes but the effective quality factor remains finite. For tunnel JJs with large R_{qp} the latter plays a minor role in THz impedance matching [24].

Nevertheless, we agree with the Reviewer that reaching $NEP \sim zW/Hz^{1/2}$ is a serious challenge. We also fully agree that just reducing I_c without changing other parameters is not possible and the issue with preserving the impedance matching could be one of the limitations. Therefore, to avoid wishful thinking, we removed the “zW” number from the manuscript and rewrote the statement as:

“Thus, we anticipate that the ultimate NEP is achieved at $T=T_{\text{MQT}}$ and $I_{c0} \sim k_B T_{\text{MQT}}/\Phi_0$. For $T_{\text{MQT}}=30\text{ mK}$, $I_{c0} \sim 6\text{ nA}$ is almost four orders of magnitude smaller than $50\text{ }\mu\text{A}$ assumed in estimations above. This indicates potential for a drastic reduction of NEP, provided the impedance matching and absorption efficiency could be preserved. The ultimate limit of NEP in such quantum phase diffusion state remains an interesting question for further investigation.”